# Genome-Wide Identification of *SMXL* Gene Family in Soybean and Expression Analysis of *GmSMXLs* under Shade Stress

**DOI:** 10.3390/plants11182410

**Published:** 2022-09-15

**Authors:** Han Zhang, Li Wang, Yang Gao, Yukai Guo, Naiwen Zheng, Xiangyao Xu, Mei Xu, Wenyan Wang, Chunyan Liu, Weiguo Liu, Wenyu Yang

**Affiliations:** 1College of Agronomy, Sichuan Agricultural University, Chengdu 611130, China; 2Sichuan Engineering Research Center for Crop Strip Intercropping System, Chengdu 611130, China; 3Key Laboratory of Crop Ecophysiology and Farming System in Southwest China, Chengdu 611130, China

**Keywords:** soybean, SMXL, evolutionary analysis, tissue specificity, shade stress

## Abstract

SMXL6,7,8 are important target proteins in strigolactone (SL) signal pathway, which negatively regulate the reception and response of SL signal, and play an important role in regulating plant branching. However, there is a relative lack of research on soybean *SMXL* gene family. In this study, 31 soybean *SMXL* genes were identified by phylogenetic analysis and divided into three groups. Based on the analysis of *GmSMXL* gene’s structure and motif composition, it was found that the *GmSMXL* members in the same group were similar. The results of cis-element analysis showed that *GmSMXL* genes may regulate the growth and development of soybean by responding to hormones and environment. Based on the tissue specificity analysis and GR24 treatment, the results showed that four *GmSMXLs* in G1 group were predominantly expressed in stems, axillary buds and leaves and involved in SL signal pathway. Finally, under shading stress, the expression of four genes in G1 group was slightly different in different varieties, which may be the reason for the difference in branching ability of different varieties under shading stress. We have systematically studied the *SMXL* gene family in soybean, which may lay a foundation for the study of the function of *GmSMXL* gene in the future.

## 1. Introduction

As an important component of aboveground plant type, plant branches have a very important impact on plant light capture, photosynthesis and resource allocation, and plant branches also affect the yield and quality of crops to a great extent [1,2]. In recent years, studies have shown that strigolactones (SLs), as a new type of plant hormone, play a very important negative role in regulating the growth and development of plant branches [3]. SLs were first found in cotton root exudates as compounds that stimulate germination of parasitic weeds in the Orobanchaceae family [4]. Later, it was found that SLs could promote the symbiotic of arbuscular mycorrhizal (AM) fungi and plant roots [5,6]. Until 2008, SLs were recognized as important plant endogenous hormones, and revealed their regulatory effects on all aspects of plant growth and development, such as SLs can inhibit plant branch growth, regulate plant secondary growth, promote leaf senescence, regulate lateral root formation and root hair development, participate in legume nodulation and so on [7,8].

With the development of molecular biology, the way of SL regulating plant branching growth and development is becoming clear. Current studies have shown that SL is mainly synthesized and transported upward in plant roots to inhibit the growth and development of plant branches [8,9]. At present, it is believed that there are three types of proteins involved in its signal transduction, namely, DWARF14 (D14) protein [10], F-box protein and SMXL/D53 protein. Like other plant hormones, SL plays its role by promoting the ubiquitination of the target proteins (D53/SMXL) and their subsequent proteasome-mediated degradation [11]: first, SL binds to the receptor protein D14, on the one hand, D14 protein acts as a hydrolase to change the molecular structure of SL, on the other hand, D14 changes its own conformation, which makes D14 and hormone complex bind to the substrate protein D53/SMXL in the nucleus [10,12]. Then recruit SCF^MAX2^ complex to form SL-D14-SCF^MAX2^-D53/SMXL protein complex, and finally 26S proteasome specifically recognizes polyubiquitin D53/SMXL protein and degrades it, thus releasing its inhibition on downstream transcription factor *BRC1* [13,14,15]. SL can inhibit plant branching or tillering. If D53/SMXL protein is not degraded, SL signal transduction is blocked, and the plant has a multi-branching or tillering phenotype [13]. SMXL/D53 protein, as the final target protein of SL signal transduction pathway, negatively regulates SL signal reception and response, and plays a very important role in the regulation of plant branching [14,16].

Previous studies have shown that the functional acquired mutant *d53* produced by dominant mutation of *D53* gene is not sensitive to SL and shows dwarfing and multi-tillering phenotype [17,18]. The orthologous gene family of *D53* in *Arabidopsis thaliana* is *SMXL* gene family, with consists of eight members [13,19]. *AtSMAX1* responds to KARs signal to regulate seed germination and hypocotyl length [20,21], while *AtSMXL6*, *AtSMXL7* and *AtSMXL8* respond to SL signal to inhibit the expression of transcription factors *BRC1*, *TCP1* and *PAP1* by binding to transcriptional corepressor protein TPL and TPL-related protein (TPR), thereby regulating plant branching, leaf morphology and lateral root growth [14,22]. *AtSMXL3*, *AtSMXL4* and *AtSMXL5* did not respond to KARs or SLs [16]. In addition, it has been found that SMXL6,7,8 are not only repressors, but also transcription factors, which negatively regulates their own transcriptional expression and maintains the dynamic balance of SMXL6,7,8 proteins and SL signal response [23]. There are also studies showing that the expression levels of *SMXL6,7* are also regulated by light environment. Recent studies have shown that FHY3 and FAR1, two orthologous transcription factors essential for phytochrome A-mediated light signaling, can directly up-regulate the expression of *SMXL6,7* [24,25]. These studies show that the *SMXL* gene family not only plays an important role in plant growth and development, but also regulates plant branching in response to changes in light signals.

A large number of studies have shown that SMXL proteins have a weak similarity with heat chaperone protein ClpB and heat shock protein HSP101 [26,27]. They have double Clp-N motifs and P-ring motifs unique to the nucleoside triphosphate hydrolase superfamily, which are induced by heat stress and are known to be related to the mechanism of proteolysis [28,29]. In addition, it has been found that SMXL6,7,8 proteins also contain ethylene response factor EAR (ethylene responsive element binding factor-associated amphiphilic repression) motif and RGKT motif, which plays an important role in SL signal transduction [30,31]. Among them, EAR motif can help SMXL7 to perform its function, but through the study of EAR mutation and deletion, it is found that it is not necessary for SMXL7 to perform its function [32]. The RGKT motif plays a role in the degradation of SMXL7/D53 after the perception of SL signal. When the functional motif of RGKT is mutated or deleted, the degradation of SMXL6,7,8 will also be affected.

Soybean is not only an important grain and oil feed crop, but also an important source of plant protein and oil in the world. Under the condition of limited arable land, maize and soybean relay strip intercropping is an important measure to effectively improve land utilization and increase soybean planting area [33,34]. However, in this model, soybean symbiosis with maize before the early flowering stage is often subjected to shade stress of high crop maize, which reduces the row light intensity and the ratio of red light to far red light, resulting in a series of shade avoidance reactions, including the reduction of branches [35]. However, previous studies have shown that the branching of relay strip intercropping soybean is the key to the yield formation, so it has important scientific significance and application prospect to study the regulation of *GmSMXL* genes on soybean branching growth and development under shade conditions to shape the ideal plant type and improve the yield of relay strip intercropping soybean.

In this study, 31 *GmSMXLs* were screened and identified from soybean genome, and their sequence characteristics, phylogenetic relationships, gene structures, conservative motif compositions, chromosome distribution and synteny were systematically analyzed and the expression of *GmSMXLs* in different tissues were detected. In addition, we also analyzed the expression patterns of *GmSMXL* genes under normal light, hormone treatment and shading conditions. These results will not only help us to better understand the function of the *GmSMXL* family, but also provide a basis for genetically modified crops, especially soybeans.

## 2. Results

### 2.1. Identification of SMXL Family Members in Soybean

In this experiment, eight SMXL proteins in Arabidopsis and two D53 proteins in O. sativa were used to construct Hidden Markov Model (HMM). The soybean genome was searched with the constructed SMXL-HMM using hmmsearch (Version 3.0, Sean Eddy, Cambridge, USA), and the sequence with E Value ≤ 1 × 10^−20^ was preserved. Subsequently, the preliminary sequences were analyzed by domain analysis software SMART (http://smart.embl-heidelberg.de/; accessed on 18 November 2020), NCBI CDD (https://www.ncbi.nlm.nih.gov/Structure/cdd/wrpsb.cgi; accessed on 18 November 2020) and PFAM (http://pfam.xfam.org/; accessed on 18 November 2020), and the sequences without conserved domains of SMXL protein were deleted. Finally, 31 GmSMXL genes were obtained and renamed according to the chromosome locations (Appendix A). As is shown in Appendix A, the CDS length of GmSMXL gens from 1365 bp (GmSMXL25) to 3777 bp (GmSMXL1). The properties of GmSMXL proteins predicted by ExPaSy (http://www.expasy.org/tools/; accessed on 20 November 2020) showed that the molecular weight of GmSMXL proteins ranged from 49.93 (GmSMXL25) to 120.70 (GmSMXL21) kD. The theoretical isoelectric point size of the coding protein ranged from 5.76 (GmSMXL14) to 8.81 (GmSMXL6), and the instability coefficients of most GmSMXL proteins are greater than 40, except for GmSMXL6, GmSMXL7, GmSMXL9, GmSMXL11, GmSMXL18, GmSMXL25, GmSMXL28 and GmSMXL29, which are unstable proteins. The prediction of subcellular localization showed that seventeen GmSMXL proteins were located in the nuclear region, six in the chloroplast, four in the mitochondria and four in the cytoplasm.

### 2.2. Phylogenetic Analysis of GmSMXL Gene Family

In order to analyze and classify the evolutionary relationships of soybean SMXL proteins, we used MEGA7 (https://www.megasoftware.net/; accessed on 24 November 2020) to construct a phylogenetic tree based on the identified 31 GmSMXL proteins in this study, eight Arabidopsis SMXL proteins and two rice D53 proteins (Figure 1). According to the evolutionary relationships, they were divided into three groups, among which G3 group contained the most GmSMXL proteins (14), followed by G2 group (11) and G1 group (6). In addition, from the phylogenetic tree, it was found that the only two OsD53 and OsD53-Like proteins in O. sativa and AtSMXL6, AtSMXL7 and AtSMXL8 proteins in Arabidopsis were all distributed in G1 group, indicating that the GmSMXL proteins of G1 group were closer to their genetic relationship and their functions were more similar.

### 2.3. Gene Structures and Motif Patterns of GmSMXL Gene Members

To further explore the evolutionary relationships between GmSMXL genes, we constructed a phylogenetic tree and analyzed the gene structure and motif characteristics of GmSMXL genes (Figure 2). Through the corresponding genomic DNA sequences and annotation files, we obtained the exon-intron pattern of GmSMXL genes. As shown in Figure 2, the intron and exon distribution of each group is relatively consistent. Most of the G1 and G2 groups contain two to three introns (GmSMXL1 contains five introns). The number of introns in the G3 group is significantly more than that in the G1 and G2 groups, and most introns are more than five. We used MEME (https://meme-suite.org/meme/doc/overview.html; accessed on 8 January 2021) to predict the conservative motif of GmSMXL members (Appendix A, Figure 2). The results show that all the GmSMXL members contain two SMXL-specific domains Clp_N (motif 3 and motif 6). Each GmSMXL members have three to ten conserved motifs, of which G3 group contains the most motifs. Members belonging to the same group have a similar motif composition. In addition, some motifs appear only in specific subgroups. For example, motif 1, motif 7 and motif 9 are unique to G2 and G3 groups.

### 2.4. Chromosome Distributions of GmSMXL Gene Members

To determine the distributions of GmSMXL genes on 20 chromosomes of soybean, we used Tbtools (Version 1.098664, Chen Chengjie, Guangzhou, China) to extract and map the chromosome locations of GmSMXL genes from the soybean genome. As shown in Figure 3, 31 GmSMXL genes were unevenly distributed on 15 soybean chromosomes. There were three GmSMXL genes on chromosomes 4, 6, 13, 17 and 18, two GmSMXL genes on chromosomes 2, 5, 8, 10, 11 and 20, but only one GmSMXL gene was observed on chromosomes 1, 14, 15 and 19, and most GmSMXL genes were distributed in areas with high gene density.

### 2.5. Synteny and Evolutionary Analyses of GmSMXL Gene Members

Gene duplication is considered to be one of the driving forces for the evolution of genomes and genetic systems [36], in which tandem duplication and segmental duplication are considered to be the main evolutionary models [37,38]. In order to find out the gene duplication events of the GmSMXL gene family, we further explored the duplication event of the identified 31 GmSMXL genes (Figure 4). The results showed that there was no tandem duplication event of GmSMXL genes in soybean. By contrast, 27 segmental duplication events related to 31 GmSMXL genes were detected. The genes with tandem duplication and segmental duplication events are also closely related genetically, which may provide clues for further study of GmSMXL genes function.

To explore the evolutionary relationship of soybean SMXL gene family, we constructed comparative syntenic graphs between *Glycine max* and *Arabidopsis thaliana*, *Glycine soja* and Oryza sativa to reveal the synteny of SMXL gene members between soybean and these three representative species (Figure 5). From the results, we found that 26 GmSMXL gene members showed synteny relationships with those in *Glycine soja* (26), *Arabidopsis thaliana* (23) and Oryza sativa (16). The number of GmSMXL orthologous genes in *Glycine soja*, *Arabidopsis thaliana* and Oryza sativa were 33, 14 and 11, respectively. Compared with Monocotyledon, GmSMXL genes consists of more dicotyledonous syntenic gene pairs. The syntenic genes between soybean and other species may play an important role in elucidating the evolution of SMXL genes.

The Ka/Ks (non-synonymous substitution/synonymous substitution) ratios of GmSMXL orthologous gene pairs among three species and within soybean species were calculated (Figure 6). The results showed that all GmSMXL orthologous gene pairs showed Ka/Ks < 1. Therefore, we speculated that soybean SMXL gene family may have experienced strong purification and selection pressure in the process of evolution.

### 2.6. Cis-Element Analyses of Soybean GmSMXL Genes

In order to study the potential role of GmSMXL genes in response to various responses, we extracted and used the 2000 bp upstream regions of the GmSMXL genes to identify the cis-elements. It can be seen from Figure 7 that GmSMXL genes are rich in a variety of cis-elements, of which light-responsive elements are the most, followed by various hormone-responsive elements, including abscisic acid responsive, gibberellin responsive, MeJA responsive, auxin responsive and salicylic acid responsive. In addition, we also found the cis-elements related to the expression of meristem and endosperm, in which the elements related to the expression of meristem were found in all the GmSMXL genes in G1 group and in some GmSMXL genes in G2 and G3 groups. The promoter region of GmSMXL genes also include many elements related to stress response, such as low temperature response, defense and stress response, circadian control and MYB-related elements, indicating that the GmSMXL gene family regulates the growth and development of soybean by responding to hormones and environment.

### 2.7. Expression Profile of GmSMXLs in Different Tissues

To explore the regulatory effect of GmSMXL genes on the branching growth and development of soybean, we selected the GmSMXL genes of G1 group for further study, because they had the closest evolutionary relationship with AtSMXL6,7,8, D53 and D53-LIKE, and their sequences were similar. At the same time, we predicted their protein tertiary structures, the results showed that the tertiary structures of G1 group GmSMXL proteins were highly similar to that of AtSMXL6,7,8 in *Arabidopsis thaliana* and D53 and D53-LIKE in Oryza sativa proteins (Appendix A). Therefore, we speculate that the function of the GmSMXL proteins of G1 group may also be highly similar to them.

To study the potential function of G1 group GmSMXL genes in soybean, we investigated the expression profiles of these genes in five soybean tissues (root, stem, leaf, axillary bud, terminal bud). It can be seen from Figure 8 that the expression levels of GmSMXL genes in G1 group were generally higher in stems, in which GmSMXL3, GmSMXL16 and GmSMXL17 were the highest in stems, followed by in axillary buds. GmSMXL21 is different from other genes in soybean leaves, followed by axillary buds and stems. In summary, AtSMXL6,7,8 orthologous genes in soybean may mainly play their functions in stems and axillary buds to regulate plant growth and development.

### 2.8. Growth and Development of Soybean Branches under Different Treatments

Light can not only provide energy for plants, but also can be used as a signal to regulate plant growth and development [39]. Plants can adjust their morphology to grow better by feeling the changes of the surrounding light environment when competing with neighboring plants for light resources [40,41]. Previous studies have shown that in the relay strip intercropping system of maize and soybean, soybean is shaded by maize, which inhibits its branching growth [35]. In order to further explore the effects of shading and SL on the growth and development of soybean branches, the axillary buds of two different varieties of soybean were treated with GR24 and shading for seven days when they grew to 2 cm, and the changes of branch length were measured and recorded.

From the branching phenotype pictures and daily branch growth length of Figure 9, we can see that after GR24 treatment, the branch growth rate of the two soybean varieties slowed down, and the inhibitory effect of GR24 on N99-6 branch growth was stronger than that of ND12. From the Figure 9D, we can see that on the 7th day of treatment, the inhibitory effect of GR24 on ND12 branching was no longer significant, and the daily branch growth of ND12 was even higher than that of normal light treatment. Under the simulated shading treatment, the branches of the two varieties of soybean stopped growing.

### 2.9. Expression Patterns of GmSMXL Genes under GR24 and Shading Treatments

In order to confirm whether GmSMXL3,16,17,21 are involved in the SL signal pathway, we sprayed 20 μM GR24 on soybean axillary buds when they grew to 2 cm (V3), and the expression of GmSMXLs in axillary buds of 1 h, 3 h and 9 h after ND12 and N99-6 treatment was determined compared with that of normal light treatment (control group). Results from Figure 10 showed that the expression of four genes in G1 group were all up-regulated in varying degrees after GR24 treament, which were consistent with the results of previous studies [23,27], indicating that all the GmSMXL3,16,17,21 in G1 group were involved in SL signal transduction. Comparing the changes of GmSMXLs in ND12 and N99-6 after GR24 treatment, it was found that GmSMXLs in ND12 were more sensitive to GR24, and their changes were more significant 0–9 h after treatment. In contrast, GmSMXLs in N99-6 were less sensitive to GR24 treatment, and their expression was significantly or very significantly up-regulated at 3 h after treatment, but there was no significant difference between N99-6 and normal light at 9 h, and even GmSMXL21 expression was significantly down-regulated at 9 h compared with normal light treatment.

In order to explore how the four GmSMXL genes of G1 group regulate the growth and development of soybean branches in response to the change of light environment, we measured the expression of GmSMXL3,16,17,21 in axillary buds at 0, 1, 3, 9 h after normal light and shading treatment. The results showed that GmSMXL genes in N99-6 were more sensitive to shading treatment than ND12. As can be seen from Figure 11, within 0–3 h of shading treatment. The expression levels of four GmSMXL genes of N99-6 changed significantly compared with normal light treatment, and all of them were significantly or very significantly up-regulated at the 9th hour of shading treatment. In ND12, except GmSMXL16, the other three GmSMXL genes were not sensitive to shading stress, and they could quickly adjust their expression to the normal light treatment level, but GmSMXL16 was very sensitive to shading stress and was significantly up-regulated within 9 h after shading treatment. We speculate that the sensitivity of these genes to shading stress and the number of sensitive genes are one of the reasons for the difference in branching ability among varieties. Further study on the function of GmSMXL16 under shading stress may help to regulate the branching growth and development of relay strip intercropping soybean, which may increase the yield of soybean.

## 3. Discussion

As a downstream regulatory factor of SL, SMXL6,7,8 is an important switch of SL signaling pathway and plays an important role in regulating the growth and development of plant branches [14]. At the same time, the study found that SMXL6 and SMXL7 are also directly regulated by far-red light signals, and then regulate the growth and development of plant branches [24]. In recent years, with the rapid development of sequencing technology, the whole genome identification of SMXL members in different species has been gradually realized. However, at present, there is a lack of research on soybean SMXL gene family. In order to better understand the GmSMXL family, we used bioinformatics to analyze the phylogenetic relationships, gene structures, motif compositions, gene chromosome distribution, synteny and cis-elements analysis of GmSMXL family members. In addition, the expression patterns of GmSMXLs in different tissues and different treatments were also discussed.

In this study, 31 GmSMXL genes were screened and identified by searching soybean genome database, which exceeded the number of SMXL genes in Arabidopsis (eight members) [16] and rice (two members) [42], indicating that the number of SMXL members may be positively correlated with genome size. According to the classifications on AtSMXL and OsD53, we divided the GmSMXL proteins into three groups. The same group of SMXL members may have similar functions in different species.

The specific domains or motifs of SMXL proteins ensure the normal exercise of their functions. In this study, all GmSMXL proteins contain double Clp_N domain, G2 group GmSMXL20 contains AAA domain, except GmSMXL1 and GmSMXL25, other GmSMXLs contain AAA_5 and AAA domain. The sequence and species of motifs fluctuate with different groups, which may reflect the different biological functions of the members of GmSMXL group. In a word, the structural consistency and differences among GmSMXL members can directly or indirectly explain their functional similarities and differences.

Soybean is an ancient polyploid with a highly duplicated genome [43]. In this study, the gene replication events of the identified GmSMXLs were analyzed, and the results showed that no tandem replication events were found in the GmSMXL family, and most of the GmSMXL genes came from segmental replication. It is speculated that the event of segmental replication is the main factor in the expansion of GmSMXL gene family. In order to further explore the evolutionary relationships of SMXL gene members among different species, we analyzed the synteny of SMXL between *Glycine max* and *Arabidopsis thaliana*, *Glycine soja* and Oryza sativa. The results showed that the SMXL synteny between soybean and dicotyledonous plants was better than that of monocotyledons, so we speculated that the synteny among SMXL members may be related to the evolutionary differentiation of species.

According to studies in Arabidopsis and rice, SMXL members are widely involved in the regulation of plant growth and development and stress [14,16,44]. We found all kinds of cis-elements in the sequence of 2000 bp upstream of GmSMXL genes transcriptional initiation site. The results show that cis-elements related to plant growth and development are widespread, especially light response elements, which are present in each GmSMXL. In addition, there are a large number of hormone-related elements, meristem and endosperm expression-related elements and so on. We also detected cis-elements related to stress, such as defense and stress-responsive elements, MYB-related elements and low-temperature response elements, which provide clues for further exploration of the function of GmSMXL genes.

Previous studies have shown that AtSMXL6,7,8 and OsD53 are downstream target proteins of SL, which positively regulate the growth and development of plant branches [13,14]. In view of this, we use the GmSMXLs of G1 group which have the closest evolutionary relationship to do further research. In order to explore the expression pattern of these genes in different tissues, we extracted and analyzed the expression levels of these genes in soybean roots, stems, leaves, terminal buds and axillary buds. The results showed that four GmSMXL genes in G1 group were highly expressed in stems, leaves and axillary buds, indicating that they may play a regulatory role in the growth and development of axillary buds.

We detected the expression of four genes in G1 group after GR24 treatment to determine whether they were involved in SL signal pathway or not. The results showed that all four GmSMXL genes in G1 group were up-regulated in varying degrees within 9 h after GR24 treatment, which proved that they were all involved in SL signal pathway, which further verified the accuracy of our screening results. Then we analyzed the expression patterns of GmSMXL3,16,17,21 in different varieties and different light environments. The results showed that the expression levels of GmSMXLs were different in varieties with different branching characteristics. In the oligobranched variety N99-6, under shading stress, the expression levels of four GmSMXL genes were significantly or very significantly up-regulated 9 h after treatment, but only GmSMXL16 in ND12 was significantly up-regulated with time after shading treatment. The expression of GmSMXL3,17,21 returned to the normal light level after 9 h shading treatment. We speculate that the difference in the number of up-regulated genes may be one of the reasons for the difference in branching ability of varieties under shade conditions. Previous studies have shown that SMXL6,7,8 genes expression levels are not only affected by SLs content, but also regulated by far red-light signal and sucrose. The results of Wang Haiyang [24] show that FHY3 and FAR1, two important transcription factors essential for phytochrome A-mediated light signaling, can directly up-regulate the transcription of SMXL6 and SMXL7. In a shaded environment, far red light can lead to a decrease in the protein levels of FHY3 and FAR1, which in turn leads to a decrease in the transcripts and protein levels of SMXL6 and SMXL7, resulting in the release of SPL9/15 protein, resulting in an increase in the transcription level of their downstream gene BRC1. So as to inhibit the generation of plant branches. At the same time, other results showed that sucrose promoted the accumulation of D53 protein, the key negative regulator of SL, and antagonized the degradation of D53 protein induced by SL [45]. In the shade environment, sucrose synthesis decreased, the antagonistic effect on D53 protein degradation weakened, and D53 protein degradation accelerated, and negative feedback regulated self-expression up-regulation. It can be seen that under shading treatment, the change of SMXL6,7,8 and D53 expression levels are regulated by many factors. At present, the systematic study of soybean SMXL genes is helpful to further understand the biological function of SMXL genes. However, further verification is needed to further reveal the response mechanism of soybean SMXL genes to shading stress.

## 4. Materials and Methods

### 4.1. Identification of GmSMXL Genes in Soybean

For the identification of *SMXL* genes in soybean, we collected all known SMXL proteins in Arabidopsis and rice. Multiple sequence alignment of SMXL proteins sequences was performed and a Hidden Markov Model (HMM) was constructed with hmmbuild (version 3.0, Sean Eddy, Cambridge, USA). Then use the SMXL-HMM to hmmsearch (version 3.0, Sean Eddy, Cambridge, USA) (E Value ≤ 1 × 10^−20^) to search the soybean genome database. Soybean genome and genome annotation files were downloaded from the Phytozome v12.1 (https://phytozome.jgi.doe.gov/pz/; accessed on 10 November 2020). Subsequently, domain analysis programs SMART (http://smart.embl-heidelberg.de/; accessed on 18 November 2020), NCBI CDD (http://www.ncbi.nlm.nih.gov/Structure/cdd/wrpsb.cgi; accessed on 18 November 2020) and PFAM (http://pfam.sanger.ac.uk/; accessed on 18 November 2020) were used to check all SMXL candidate proteins sequences to remove sequences lacking double Clp-N motifs or p-loop NTPase domain. The number of amino acids, molecular weight (MW), theoretical isoelectric point (PI) and instability index of selected soybean proteins were obtained by ExPASy (https://web.expasy.org/protparam/; accessed on 20 November 2020) proteomics server. The subcellular location of GmSMXL proteins were predicted by CELO (http://cello.life.nctu.edu.tw/; accessed on 6 January 2021) online tool.

### 4.2. Sequence Alignment and Phylogenetic Analysis of GmSMXL Proteins

The SMXL protein sequences of *O. sativa* and *Arabidopsis thaliana* were downloaded from Uniprot (https://www.uniprot.org/; accessed on 24 November 2020) and compared with GmSMXLs with the default parameters of ClustalW in MEGA7.0 (https://www.megasoftware.net/; accessed on 24 November 2020), and then the evolutionary tree was constructed by neighbor-joining (NJ). The parameters were set as follows: bootstraps repeat times 1000, Poisson model, partial deletion of gap. The use the Evolview (https://evolgenius.info//evolview-v2/#login; accessed on 24 November 2020) online website to beautify the evolutionary tree.

### 4.3. Gene Structures and Covered Motif Analyses of GmSMXLs

The program MEME (https://meme-suite.org/meme/doc/overview.html; accessed on 8 January 2021) was used to determine the conserved motifs among GmSMXLs, the maximum number of motifs was set to 10, and all other parameters were set to default. Then use the TBtools (Version 1.098664, Chen Chengjie, Guangzhou, China) to analyze the *GmSMXL* genes structures and visualize the evolutionary tree, gene structure and MEME analysis results.

### 4.4. Chromosome Locations, Duplications and Synteny Analysis of GmSMXL Genes

According to the soybean genome information available on Phytozome, the chromosome locations and duplications of *GmSMXL* genes were mapped, and the gene density information of each chromosome was calculated by TBtools and displayed by TBtools software. In order to explore the synteny relationships of *SMXL* genes between soybean and other species, we downloaded genome data and gene annotation files of *Arabidopsis thaliana* (Tair Annotation Release 10), *Glycine Soja* (v1.1) and *Oryza Sativa* (MSU Annotation Release 7.0) from Phytozome. The syntenic analyzing graphs were constructed by using the Dual Synteny Plotter function in TBtools. Non-synonymous substitution (Ka) and synonymous substitution (Ks) of duplicated *SMXL* genes were calculated by TBtools.

### 4.5. Analysis of Cis-Elements in the Promoter Region of GmSMXL Genes

The upstream 2000 bp sequences of each *GmSMXL* genes were extracted by TBtools and submitted to the online website PlantCARE (http://bioinformatics.psb.ugent.be/webtools/plantcare/html/; accessed on 2 December 2020) to predict the cis-acting elements in the promoter regions. The diagram of cis-acting elements of *GmSMXL* genes was displayed by TBtools.

### 4.6. Prediction of Tertiary Structure of GmSMXL Proteins

The tertiary structure of GmSMXL proteins were predicted by SWISS-MODEL (https://swissmodel.expasy.org/; accessed on 20 May 2021) and Phyre2 (http://www.sbg.bio.ic.ac.uk/phyre2/html/page.cgi?id=index; accessed on 3 March 2021). Combining the prediction results of the two online tools, Phyre2 prediction has high reliability and can be used as a reference for the structure of GmSMXL.

### 4.7. Plant Materials, Growth Conditions and Shade Treatment

The soybean materials selected in this experiment were multi-branched Nandou 12 (ND12) and oligobranched Nannong 99-6 (N99-6). Among them, ND12 was bred by Agricultural Science Research Institute of Nanchong City, Sichuan Province, China, and N99-6 was bred by Nanjing Agricultural University. Soybean plants grew under the conditions of relative humidity 60%, light 12 h (25 °C)/12 h (25 °C), light intensity 680 μ mol m^−2^s^−1^. In order to analyze the expression patterns of *GmSMXLs* in different tissues of soybean, we collected roots, stems, leaves, apical meristems and axillary buds of ND12 and N99-6 at V3 stages. In addition, in order to explore the change trend of *GmSMXLs* expression under GR24 and shading treatment and the growth and development of soybean axillary buds, we sprayed soybean axillary buds at V3 stage with 20 μM GR24 and simulated shading treatment with green film (light intensity 120 μmol m^−2^s^−1^, red light/far red light R/FR 0.6). Finally, the soybean axillary buds were obtained at 0, 1, 3 and 9 h after treatment and stored in the refrigerator at −80 °C. The rest of the plants were treated for 7 days, and the development of axillary buds was observed.

### 4.8. Quantitative RT-PCR (qRT-PCR) for GmSMXL Genes

The total RNA was extracted by using the Plant RNA Kit (OMEGA) from different parts of soybean and axillary buds after different treatments. The Tli RNaseH Plus (TaKaRa, Beijing, China) was used to synthesize first-strand cDNA. The qRT-PCR was performed using Vazyme™ AceQ qPCR SYBR Green Master mix (TaKaRa, Beijing, China) on a QuantStudio 6 Flex Real-Time PCR System (Thermo Fisher Scientific, Waltham, Massachusetts, USA). The housekeeping *GmActin* gene was determined as an internal control. Triplicate quantitative assays were performed on each cDNA sample and analyzed by a 2^−^^ΔΔCT^ method.

### 4.9. Statitical Analysis

In this study, the Student’s *t*-test in Graphpad Prism 8 (Version 8.0, Motulsky, San Diego, State of California)was used to analyze the differences between different treatments. *p*-value cut-off of 0.05 was considered as statistical significance. All the error bars were standard error (SE).

## 5. Conclusions

In this study, we identified 31 *GmSMXL* genes in soybean. These *GmSMXL* genes were unevenly distributed on 15 chromosomes of soybean and were divided into three groups according to their phylogenetic relationship. Further analysis of their gene structures and motif compositions showed that the *SMXL* members in the same group were widely similar, which may indicate that their gene functions are similar. In addition, synteny analyses showed that the emergence of new *GmSMXL* genes were mainly dominated by segmental replication events. Cis-element and gene expression analysis revealed the potential regulation of identified *GmSMXL* genes in different tissues, GR24 treatment and shading stress. It was proved that *GmSMXL3,16,17,21* in G1 group were involved in the regulation of soybean branching growth and development under shading stress, and their expression patterns were different in different branching ability varieties. To sum up, we comprehensively studied the characteristics of soybean *SMXL* genes, which provided valuable clues for understanding the biological function of *GmSMXL* genes and further studying *GmSMXL* genes.

## Figures and Tables

**Figure 1 plants-11-02410-f001:**
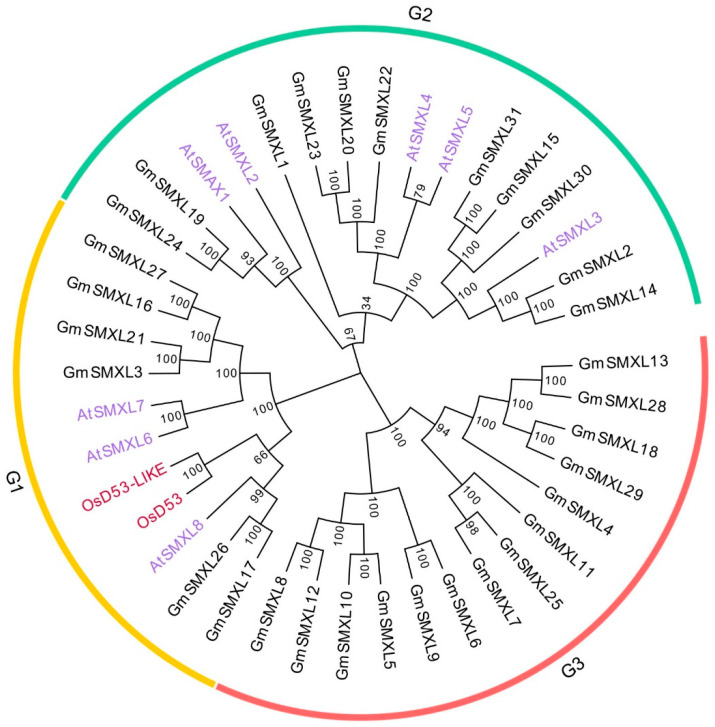
Phylogenetic tree of SMXL proteins from *Glycine max*, *Arabidopsis thaliana* and Oryza sativa. The SMXL protein sequences of the three species were aligned by the ClustalW method implemented in the MEGA version7, and the tree was built with the neighbor-joining (NJ) method. Gm represents *Glycine max*; At represents *Arabidopsis thaliana*; Os represents *Oryza sativa*, respectively.

**Figure 2 plants-11-02410-f002:**
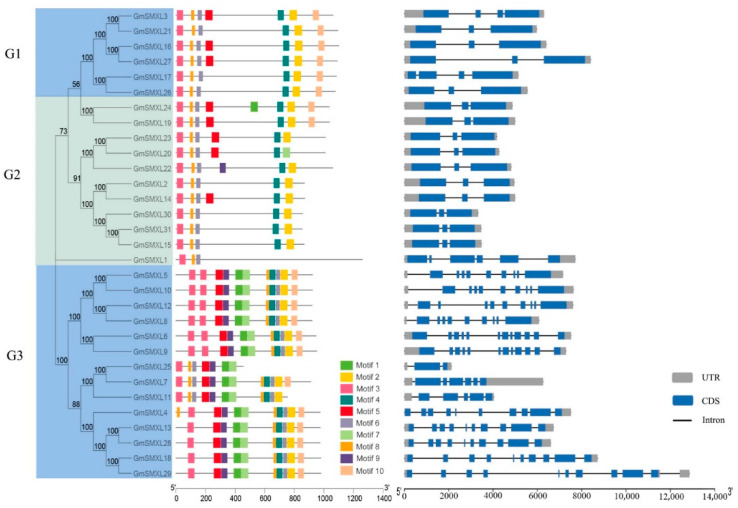
Phylogenetic analysis, gene structures and conserved motifs of GmSMXLs. The three columns in the diagram are phylogenetic tree, motif, and exon-intron in order from left to right. The sequence information for each MEME-motif was provided in Appendix A.

**Figure 3 plants-11-02410-f003:**
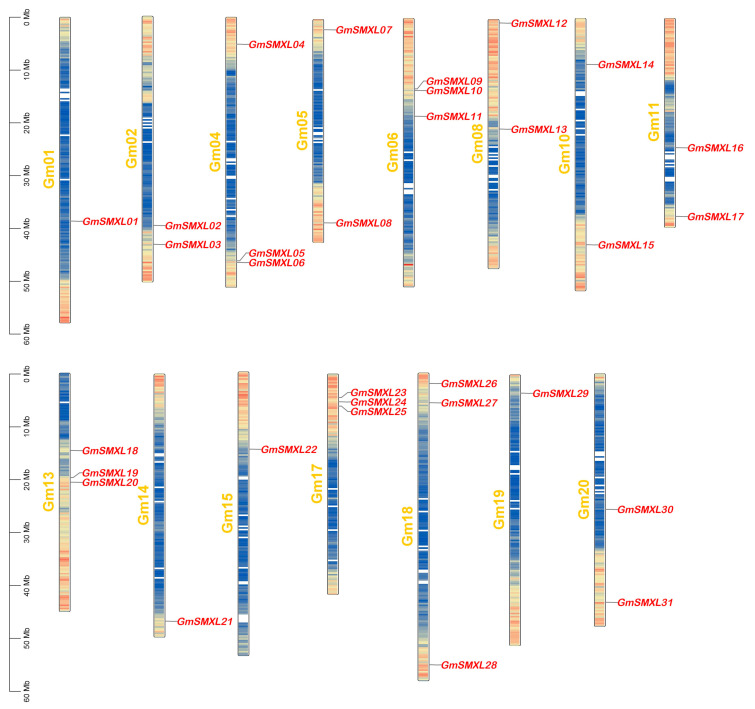
Chromosomal distributions of GmSMXL genes. The names of the SMXL genes are labeled at the appropriate position on the right side of each soybean chromosome. The gene density of each chromosome was illustrated by gradient colors, blue represents low gene density, red represents high gene density, and the genetic interval was set to 300 kb for evaluation.

**Figure 4 plants-11-02410-f004:**
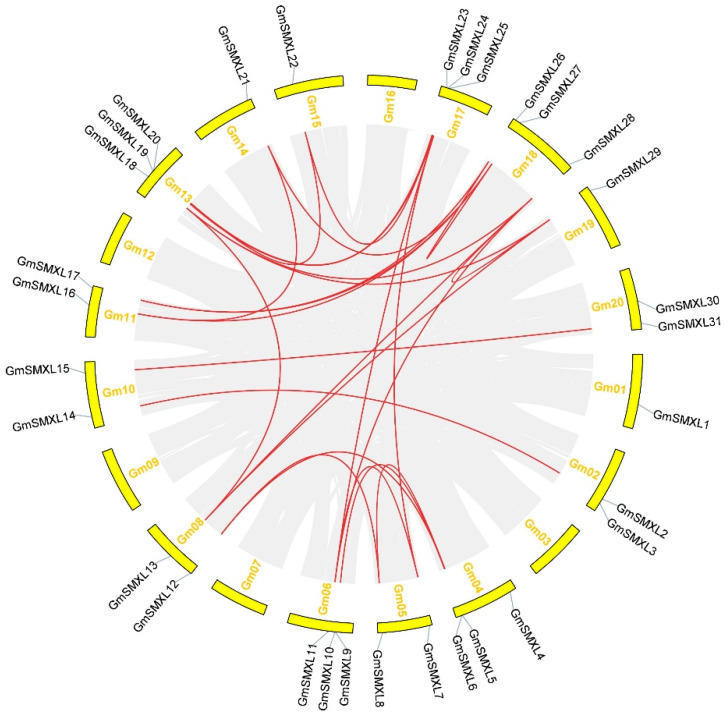
Collinearity analysis of the whole soybean genome and SMXL family genes. The gray lines in the middle represent the soybean genome’s collinearity module, and the red lines show that some of the GmSMXL genes have a collinear relationship.

**Figure 5 plants-11-02410-f005:**
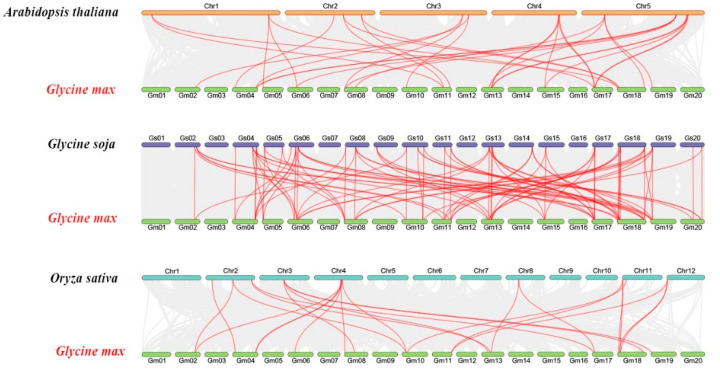
Synteny analyses of the SMXL genes between soybean and three representative species. The collinear blocks within soybean and other specie genomes were displayed by the gray lines. The syntenic SMXL gene pairs between soybean and other species were highlighted with the red lines.

**Figure 6 plants-11-02410-f006:**
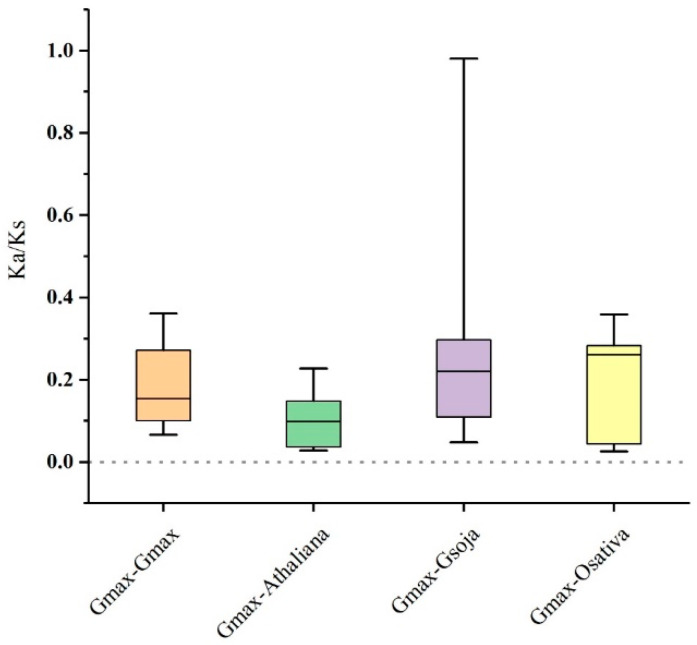
The ratio of nonsynonymous to synonymous substitutions (Ka/Ks) of SMXL genes in soybean and other three species. The species’ names with the prefixes’Gmax’, ‘Athaliana’, ‘Gsoja’ and ‘Osativa’ indicated *Glycine max*, *Arabidopsis thaliana*, *Glycine soja* and *Oryz sativa*, respectively.

**Figure 7 plants-11-02410-f007:**
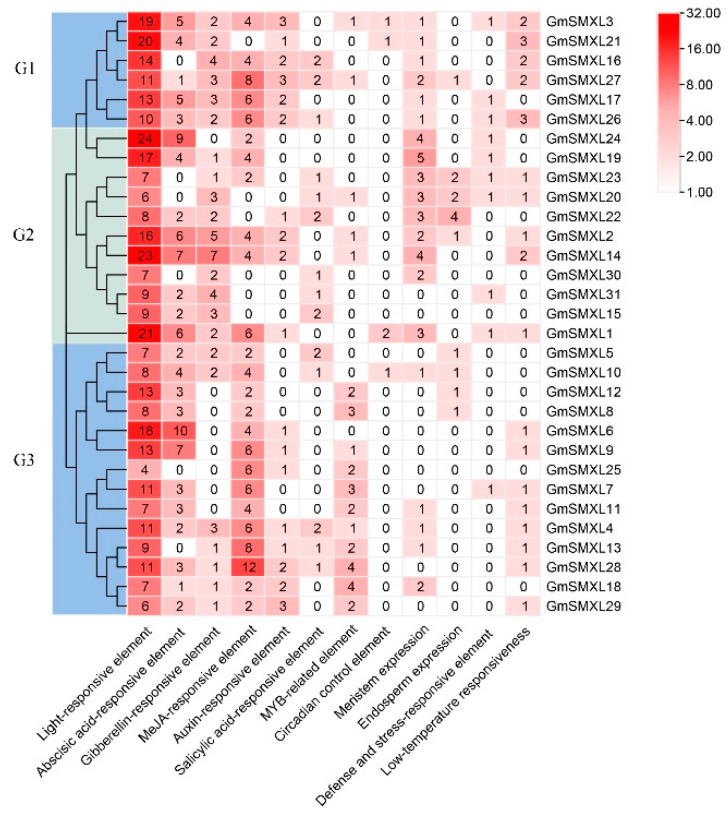
Cis-elements in the GmSMXL genes promoter regions. The number of cis-elements is displayed by the heat map. The redder the color, the more the number of cis-elements. The number represents the specific number of cis-elements of the gene.

**Figure 8 plants-11-02410-f008:**
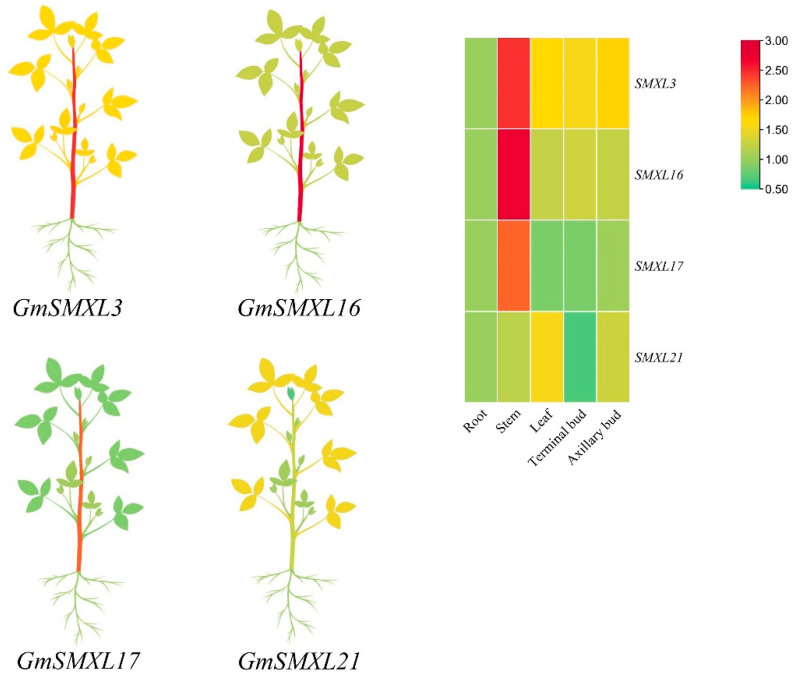
The differential expression of representative GmSMXL genes in five tissues by RT-qPCR. The mean expression value was visualized by Tbtools; red represents high expression level and green represents low expression level.

**Figure 9 plants-11-02410-f009:**
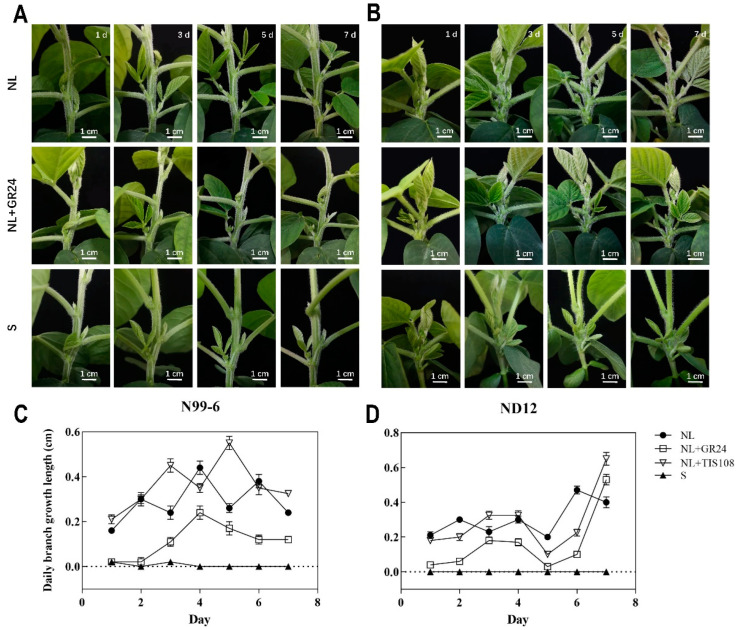
Changes of branching length of ND12 and N99-6 under normal light (NL), normal light spraying GR24 (NL+GR24) and shading conditions (S) for one week. (**A**,**B**) Changes in the length of axillary buds of N99-6 and ND12 under normal light, GR24 and shade treatment on 1/3/5/7 day. The size of the soybean branches can be estimated by using the 1 cm scale at the right bottom. (**C**,**D**) The average daily branch growth length of N99-6 and ND12 in 7 days after normal light, GR24 and shade treatment. Error bars represent standard errors.

**Figure 10 plants-11-02410-f010:**
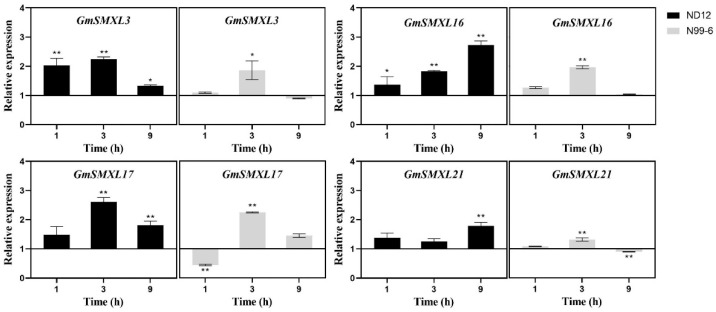
The expression levels of GmSMXL genes in ND12 and N99-6 relative to their normal light treatment after 1, 3, 9 h GR24 (20 μM) treatment. The values referred to the mean ± standard error (SE) of three independent biological replicates. Asterisks manifested the corresponding genes significantly up- or down-regulated compared with those in normal light treatment. (* *p* < 0.05, ** *p* < 0.01, Student’s *t*-test).

**Figure 11 plants-11-02410-f011:**
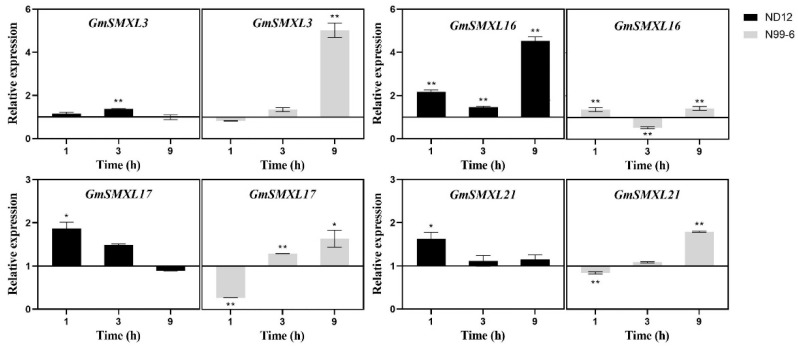
The expression levels of GmSMXL genes in ND12 and N99-6 relative to their normal light treatment after shading treatment. The values referred to the mean ± standard error (SE) of three independent biological replicates. Asterisks manifested the corresponding genes significantly up- or down-regulated compared with those in normal light treatment. (* *p* < 0.05, ** *p* < 0.01, Student’s *t*-test).

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
