# Peer review of "Genome-Wide Identification of SMXL Gene Family in Soybean and Expression Analysis of GmSMXLs under Shade Stress"

_plants, 2022, doi:10.3390/plants11182410_

Round 1
Reviewer 1 Report
In this study, authors characterized 32 soybean SMXL genes by bioinformatic analyses and conducted gene expression analysis of identified GmSMXL genes under shade stress. The content and purpose of this study are fine for publication. The in silico data analyses were nicely done with detailed methods. They found that the four identified genes under shading stress might be associated with branching ability under shading stress.
Check the title and subtitle format according to Plants. All nouns should not be capitalized.
The abstract is too long. The length of abstract should be less than 200 words.
In addition, please rewrite the abstract concisely. The first sentence in the abstract was too long.
The conclusions seem to be an abstract.
Introduction was nicely written but was also too long. Authors described too many SMLX genes. For example, the paragraph (L68-L91) can be reduced.
In introduction, it would be very nice to mention the main reason to study soybean GmSMXL genes in this study.
great extent[1, 2]. -> great extent [1, 2]. A space between a sentence and references. Check them thoroughly in the manuscript.
L124 8 SMXL proteins in Arabidopsis and 2 D53 proteins -> eight SMXL proteins in Arabidopsis and two D53 proteins
L131-L133 This should be description for Table S1 not in the main manuscript.
L134 1365bp -> 1365 bp (space between number and an unit)
L142-143 six in the chloroplast, four in the mitochondria and four in the cytoplasm. Check them thoroughly in the manuscript.
L156 aligned by the ClustalW method implemented in the MEGA version 7.0.
L157 It would be much better to generate the phylogenetic tree using maximum likelihood method with bootstrap values.
L163 figure 3 -> Figure 3, in addition, this should be Figure 2. Figure should be capitalized. Check them thoroughly in the manuscript.
L169 figure 2 -> Figure 3, Please reorganize the order of Figure 2 and Figure 3.
L441-443 Please construct the phylogenetic tree using maximum likelihood method with bootstrap values.
L458 Scientific names should be italicized.
I recommend authors to review their manuscript carefully for revision.
Overall, this manuscript is very good for publication after revision.
Reviewer 2 Report
The manuscript by Zhang et. Al. entitled “Genome-Wide Identification of SMXL Gene Family in soybean and expression analysis of GmSMXLs under Shade stress” reports that the 31 soybean SMXL genes were screened and identified by bioinformatics analysis. Four genes expression in G1 group investigated by RT-qPCR shows slightly different between N99-6 and ND12. The authors claim that they ‘systematically studied the SMXL gene family in soybean, which may lay a foundation for the study of the function of GmSMXL gene in the future’. The work is a kind of technically sound piece of research, and within the scope of Plants, but the weight of it is questionable. It requires major revision before its acceptance for publication.
1. Provide the qPCR data of the other two groups
2. Provide qPCR data from second bench ‘shade’ treatment
3. Provide acc num and primer seq of the GmActin
4. Provide more references related to GmSMXLs
5. The statistical method ‘T-Test’ is not goodness of fit for the data analysis of this research.
Round 2
Reviewer 2 Report
The Fig. 10 and 11 must be rearranged if the authors keep sticking on the T-TestAuthor Response
Please see the attachment.
